# D-Dimer Level and Neutrophils Count as Predictive and Prognostic Factors of Pulmonary Embolism in Severe Non-ICU COVID-19 Patients

**DOI:** 10.3390/v13050758

**Published:** 2021-04-26

**Authors:** Benjamin Thoreau, Joris Galland, Maxime Delrue, Marie Neuwirth, Alain Stepanian, Anthony Chauvin, Azeddine Dellal, Olivier Nallet, Melanie Roriz, Mathilde Devaux, Jonathan London, Gonzague Martin-Lecamp, Antoine Froissart, Nouara Arab, Bertrand Ferron, Marie-Helene Groff, Viviane Queyrel, Christine Lorut, Lucile Regard, Emilie Berthoux, Guillaume Bayer, Chloe Comarmond, Bertrand Lioger, Arsène Mekinian, Tali-Anne Szwebel, Thomas Sené, Blanca Amador-Borrero, Olivier Mangin, Pierre O. Sellier, Virginie Siguret, Stéphane Mouly, Jean-Philippe Kevorkian, Dominique Vodovar, Damien Sene

**Affiliations:** 1Department of Internal Medicine, National Referral Center for Rare Systemic Autoimmune Diseases, Cochin Hospital, AP-HP, University of Paris, CEDEX 14, 75679 Paris, France; tali-anne.szwebel@aphp.fr; 2INSERM U1016, Cochin Institute, Paris, University of Paris, CNRS UMR 8104, 75014 Paris, France; 3Department of Internal Medicine, Lariboisière Hospital, Assistance Publique-Hôpitaux de Paris, (AP-HP) Université de Paris, 75010 Paris, France; jorisgalland@icloud.com (J.G.); Blanca.AMADOR-BORRERO@aphp.fr (B.A.-B.); oli_mangin@hotmail.fr (O.M.); stephane.mouly@aphp.fr (S.M.); damien.sene@aphp.fr (D.S.); 4Haemostasis Laboratory, Lariboisière Hospital, AP-HP, University of Paris, 75010 Paris, France; maxime.delrue@aphp.fr (M.D.); marie.neuwirth@aphp.fr (M.N.); alain.stepanian@aphp.fr (A.S.); virginie.siguret@aphp.fr (V.S.); 5Emergency Department, Lariboisière Hospital, AP-HP, University of Paris, 75010 Paris, France; anthony.chauvin@aphp.fr; 6Department of Rheumatology and Internal Medicine, Le Raincy-Montfermeil Hospital, 93370 Montfermeil, France; azeddine.dellal@ght-gpne.fr; 7Department of Cardiology, Le Raincy-Montfermeil Hospital, 93370 Montfermeil, France; olivier.nallet@ght-gpne.fr; 8Department of Internal Medicine, Hospital Center of Agen, 47923 Agen, France; rorizm@ch-agen-nerac.fr; 9Department of Internal Medicine, Hospital Center of Poissy-Saint Germain, 78300 Saint Germain en Laye, France; mathilde.devaux@ght-yvelinesnord.fr; 10Department of Internal Medicine, Diaconesses Croix Saint-Simon Hospital, 75012 Paris, France; JLondon@hopital-dcss.org; 11Department of Internal Medicine, Hospital Center of Pau, 64000 Pau, France; g.martinlecamp@gmail.com; 12Department of Internal Medicine, Intermunicipal Hospital Center of Créteil, 94000 Créteil, France; Antoine.Froissart@chicreteil.fr (A.F.); nouara.Arab@chicreteil.fr (N.A.); 13Department of Internal Medicine, Hospital Center of Sens, 89100 Sens, France; bferron@ch-sens.fr; 14Department of Internal Medicine, Hospital Center of Nord-Mayenne, 53100 Mayenne, France; mgroff@ch-mayenne.fr; 15Department of Rheumatology, University Hospital of Nice, 06000 Nice, France; queyrel-moranne.v@chu-nice.fr; 16Department of Pneumology, Cochin Hospital, AP-HP, Université de Paris, 75014 Paris, France; christine.lorut@aphp.fr (C.L.); lucile.regard@aphp.fr (L.R.); 17Department of Internal Medicine, Saint Luc-Saint Joseph Hospital, 69007 Lyon, France; emilie.berthoux@gmail.com; 18Department of Internal Medicine, Claude Galien Hospital, 91480 Quincy sous Senart, France; guillaume.bayer@hotmail.fr; 19Department of Internal Medicine, Pitié-Salpétrière Hospital, AP-HP, Sorbonne University, 75013 Paris, France; chloe.comarmondortoli@aphp.fr; 20Department of Internal Medicine, Simone Veil Hospital, 41000 Blois, France; liogerb@ch-blois.fr; 21Department of Internal Medicine, Saint Antoine Hospital, APHP, 75012 Paris, France; arsene.mekinian@aphp.fr; 22Department of Internal Medicine, Fondation Rothschild, 75019 Paris, France; thomas_sene1@yahoo.fr; 23Department of Infectious Disease, Lariboisière Hospital, APHP, 75010 Paris, France; pierre.sellier@aphp.fr; 24Department of Endocrinology, Lariboisière Hospital, APHP, 75010 Paris, France; jean-philippe.kevorkian@aphp.fr; 25Centre Anti-Poison, Fernand Widal Hospital, AP-HP, University of Paris, 75010 Paris, France; dominique.vodovar@aphp.fr; 26INSERM UMRS 1144, 75006 Paris, France

**Keywords:** COVID-19, pulmonary embolism, D-dimer, neutrophil, anticoagulation, predictive factor, prognostic, mortality, ICU transfer

## Abstract

The incidence of pulmonary embolism (PE) is high during severe Coronavirus Disease 2019 (COVID-19). We aimed to identify predictive and prognostic factors of PE in non-ICU hospitalized COVID-19 patients. In the retrospective multicenter observational CLOTVID cohort, we enrolled patients with confirmed RT-PCR COVID-19 who were hospitalized in a medicine ward and also underwent a CT pulmonary angiography for a PE suspicion. Baseline data, laboratory biomarkers, treatments, and outcomes were collected. Predictive and prognostics factors of PE were identified by using logistic multivariate and by Cox regression models, respectively. A total of 174 patients were enrolled, among whom 86 (median [IQR] age of 66 years [55–77]) had post-admission PE suspicion, with 30/86 (34.9%) PE being confirmed. PE occurrence was independently associated with the lack of long-term anticoagulation or thromboprophylaxis (OR [95%CI], 72.3 [3.6–4384.8]) D-dimers ≥ 2000 ng/mL (26.3 [4.1–537.8]) and neutrophils ≥ 7.0 G/L (5.8 [1.4–29.5]). The presence of these two biomarkers was associated with a higher risk of PE (*p* = 0.0002) and death or ICU transfer (HR [95%CI], 12.9 [2.5–67.8], *p* < 0.01). In hospitalized non-ICU severe COVID-19 patients with clinical PE suspicion, the lack of anticoagulation, D-dimers ≥ 2000 ng/mL, neutrophils ≥ 7.0 G/L, and these two biomarkers combined might be useful predictive markers of PE and prognosis, respectively.

## 1. Introduction

Respiratory impairment secondary to Coronavirus Disease 2019 (COVID-19), due to the SARS-CoV-2 virus infection, is responsible for severe clinical patterns that are associated with higher morbidity and mortality [1]. While COVID-19 by itself can cause severe pneumonia, high prevalence of venous thromboembolism (VTE) has been reported, including onset of pulmonary embolism (PE) that impacts the COVID-19 prognosis [2,3]. The underlying pathophysiological mechanisms of COVID-19-related coagulopathy are multifactorial, involving a hypercoagulability state, fibrinolysis defect, an endotheliopathy, leading to the immunothrombosis. [4,5,6]. Depending on both the series and the hospitalization setting (intensive care unit ICU vs. Non-ICU department), the incidence of PE ranges from 3% to 35% in COVID-19 patients [2]. Several risk factors have been reported as predictive of PE in those patients. The most frequently reported predictive factor is D-dimer blood concentration, with cut-off values ranging from 1000 to 3000 ng/mL [7,8,9], with questionable predictive power if used alone, thus leading to identify others [10,11,12].

PE might be inaugural and diagnosed upon admission in emergency departments (ED), and we have previously identified D-dimer levels and laboratory of inflammatory response (white blood count (WBC) and ferritin levels) as predictive factors for the early detection of PE upon ED admission in preliminary data from the CLOTVID cohort. However, worsening of the respiratory condition and PE might also occur after the transfer from ED to medical wards.

As the management of the thromboembolic risk during the COVID-19 (i.e., anticoagulant prophylaxis and screening process of PE) is not consensual, the identification of clinical, laboratory, or CT-scan features to predict PE onset and prognosis in COVID-19 patients remains paramount in non-ICU hospitalized COVID-19 patients. We believe that identifying factors associated and/or predictive of PE occurrence and poor prognosis might be useful to clinicians.

In the current study, we therefore aimed to identify clinical, laboratory, and CT-scan predictive factors of PE occurrence as well as prognostic factors for death and/or transfer to ICU in non-ICU COVID-19 patients hospitalized during more than 24 h.

## 2. Methods

### 2.1. Setting

Enrolled patients belonged to the multicenter (18 participating French hospitals) observational French CLOTVID cohort that retrospectively collected data from non ICU COVID-19 hospitalized patients from 6 April to 28 April 2020. 

### 2.2. Patients and Inclusion Criteria

Considered for study were all consecutive adults (≥18 years old) with confirmed COVID-19 infection (positive RT-PCR) hospitalized in a participating Internal Medicine and Pneumology ward and for whom an in-hospital CT pulmonary angiography (CTPA) was performed for clinical suspicion of PE based on the presence of chest pain, tachycardia, electrocardiogram abnormality, dyspnea, worsening of SpO2 or an increase of oxygen requirements, lower limb pain, D-dimer levels (upon admission or its evolution), and the absence of respiratory improvement. Patients diagnosed with an isolated deep venous thrombosis (DVT) were excluded, as were patients directly admitted to the ICU without initial hospitalization in general ward (ICU patients). The whole population was divided into two subsets depending on whether the CTPA was made in the ED or during hospitalization in a medicine ward to avoid confounding bias related to thromboprophylaxis and its consequences on laboratory markers. This study analyzed COVID-19 patients with a confirmed PE (PE subset) or not (NON-PE subset) based on CTPA performed during the medicine ward stay (at least 24 h after transfer from ED, Figure 1).

All patients gave their informed consent to participate, and all data were recorded through a standardized clinical report form (CRF). This study was approved by the Institutional data protection authority of Assistance Publique-Hôpitaux de Paris (University hospital of Paris), reference 2,217,565 v0; 12 April 2020.

### 2.3. Data Collection

Demographic data (age, gender, body mass index, comorbidities) and past medical history with risk factors of severe form of COVID-19 or pro-thrombotic risk disease and previous long-term anticoagulation therapy were recorded. COVID-19 infection history (onset date and clinical presentation), COVID-19-related lung injury based on admission CT-scan, i.e., ground glass or condensation, extension of lesions: absent (<10%), minimal (10–25%), moderate (25–50%), extensive (50–75%), severe (>75%) [13], laboratory tests upon admission and at the PE suspicion time (white blood count, serum creatinine level, C-reactive protein, serum ferritin, brain natriuretic peptide and troponin, prothrombin time ratio, fibrinogen, plasma D-dimers), oxygen flow, symptoms and vital parameters at the PE suspicion time, use of a thromboprophylaxis therapy and dose adapted on BMI (high dose) or not (standard dose), use of available medications to treat COVID 19-infection (standard of care, antiviral therapies (lopinavir/ritonavir or hydroxychloroquine/chloroquine), steroids, biotherapies), and outcome (death, ICU admission, and recovery) were also collected in all patients. The Wells score associated with D-dimer threshold adjusted on age was retrospectively determined. Each CTPA was locally reviewed by a radiologist blinded to the hypothesis and clinical status. 

### 2.4. Statistics

Data were reported as numbers (n) and percentages (%) for categorial variables and compared by using the Chi-square or Fisher’s exact test, as appropriate. Continuous variables were expressed as median and interquartile range (IQR) and were compared by using the Mann–Whitney U-test. Data were compared between the two subsets according to the CTPA conclusion concerning the PE diagnosis (PE vs. NON-PE group) in the post-admission population. 

To explore the risk factors associated with PE onset, univariate and multivariate logistic regression models were used with results expressed by odds ratio (OR) with their respective 95% confidence interval (95%CI). To assess the predictive value of D-dimer and neutrophils for PE occurrence, we analyzed respective receiver operating characteristics (ROC) curve and assessed area under the curve (AUC), sensitivity and specificity, positive and negative predictive values, as well as positive and negative likelihood ratio of the optimal cut-offs (Appendix A
Appendix A). Correlation between D-dimer levels upon admission and level at the suspicion time of PE was assessed using Spearman coefficient. 

The impact of potential predictors on the composite risk of transfer to the ICU or death, whichever came first vs. hospitalization discharge, were assessed using Cox regression models with hazard ratio (HR) with 95% CI calculation and log-rank test. The PE status (presence or not) and some biomarkers were assessed. 

Data were analyzed using R software, v3.6.1 (R Foundation for Statistical Computing, Vienna, Austria; http://www.R-project.org/; accessed on 25 April 2021). The study was built and results were reported according to the guidelines on the Strengthening the Reporting of Observational Studies in Epidemiology (STROBE) [14].

## 3. Results

### 3.1. Patients’ Characteristics

Among the 174 patients with suspected PE, PE was suspected during the medicine ward hospitalization (i.e., post admission) in 86 patients (Figure 1). Among these 86 patients (50/86 males (58.1%), median (IQR) age 66 years [55–77]) enrolled with a post admission PE suspicion, 30 patients (35%) had confirmed PE and 56 (65%) had not (NON-PE). Baseline characteristics are presented in Table 1. In PE patients, thrombus localization was proximal in 6 patients (20%), purely segmental in 19 (63.3%), sub-segmental in 5 (16.7%), and mainly unilateral (18/30–60%). Men were over-represented in the PE group (23/30, 76.7% vs. 27/56, 48.2%, *p* = 0.01), while no significant differences were observed between PE and NON-PE groups regarding comorbidities, number of previous thrombotic risk factor, long-term anticoagulation, and COVID-19 clinical presentation (Table 1).

### 3.2. Clinical Manifestations

No difference was found in clinical symptoms between PE and non-PE (*p* > 0.05 for each symptom compared). Time from first COVID-19 attributable symptom to CTPA run and time from admission to PE suspicion (15 [10,11,12,13,14,15,16,17,18] and 6 days [4,5,6,7,8], respectively) did not differ between groups (*p* > 0.05). In contrast, we observed significant differences in laboratory parameters between the two groups. The PE group displayed higher median D-dimer levels (9710 ng/mL [IQR 3310–20,000] vs. 1580 [863–2972]; *p* < 0.0001) and a greater proportion of patients with D-dimer ≥ 2000 ng/mL (76.2% vs. 30.4%, *p* < 0.0001), higher white blood count (WBC 8.9 G/L [6.7–11.9] vs. 6.7 [4.7–8.4], *p* = 0.002), and higher neutrophil count (7.0 vs. 5.4 G/L; *p* = 0.03) (Table 1). The neutrophil-lymphocyte count ratio was higher in patients with D-dimer ≥ 2000 ng/mL compared to those without (7.8 vs. 3.8, *p* = 0.006). Upon admission, patients with PE had higher WBC (9.4 vs. 6.8 G/L, *p* < 0.01) and D-dimer level (4335 vs. 1343 ng/mL, *p* < 0.001). Prothrombin time ratio was slightly lower in patients with PE (*p* = 0.04, Appendix A).

### 3.3. Treatment and Outcomes

The detailed treatment of patients is listed in Table 2. The proportion of patients in PE group previously treated with anticoagulant (long term of thromboprophylaxis) was lower in the PE than in NON-PE subset (80.0% vs. 96.4%, *p* = 0.02). Four patients previously treated with long-term anticoagulant also received thromboprophylaxis after discontinuation, including three in the PE group.

Patients with PE were more frequently transferred to the ICU (6/30, 20.0% vs. 1/56, 1.8%, *p* < 0.01) and conversely tended to be less frequently discharged from hospitalization (46.7 vs. 67.9%, *p* = 0.05). No difference was noted regarding death between the two groups (Table 2). However, patients who died had a higher level of D-dimers compared with those who survived (median 15,250 ng/mL [9745–20,000] vs. 2190 [1370–7990], *p* = 0.01).

### 3.4. Predictive Factors Associated with Pulmonary Embolism 

Clinical and laboratory parameters that significantly differed between the two groups at the time of PE suspicion were included into a logistic regression model. The identified thresholds through the ROC curves were used for selected biomarkers (Appendix A
Appendix A). Univariate analysis showed that male gender (*p* = 0.01), absence of long-term anticoagulant or thromboprophylaxis (*p* = 0.02), D-dimer level ≥ 2000 ng/mL (*p* = 0.006), and neutrophils count ≥ 7.0 G/L (*p* = 0.01) were significantly associated with the diagnosis of PE. Using multivariate model, absence of anticoagulant (OR [95%CI], 72.3 [3.6–4384.8], *p* = 0.01) as well as D-dimer levels ≥ 2000 ng/mL (26.3 [4.1–537.8], *p* = 0.004), and neutrophils count ≥ 7.0 G/L (5.8 [1.4–29.5], *p* = 0.02) were independently associated with risk of PE (Table 3). All diagnostic parameters of potential predictive variables based on ROC curve are displayed in Table 4 and Appendix A
Appendix A. Considering biomarker levels upon admission, only the D-dimer levels remained independently associated with the PE occurrence (*p* = 0.01). The optimal threshold identified by ROC curve was 1700 ng/mL (AUC 0.80 [95%CI 0.66–0.93], sensitivity and specificity of 88.9% and 63.4% respectively, data not shown). D-dimers upon admission and at the suspicion time were strongly correlated (R^2^ = 0.63, *p* < 0.0001, Appendix A
Appendix A). The Wells score had very weak performance parameters (sensitivity 0.13 [0.04–0.31] and negative predictive value 0.64 [0.52–0.75]). Only 4/86 patients (4.7%) with confirmed PE had a positive value. Patients with a PE occurrence displayed a negative Wells score in 30.2% (26/86). In contrast, at the suspicion time of PE, the composite criterion that included D-dimer level ≥ 2000 ng/mL and neutrophils count ≥ 7.0 G/L was associated with a 15-fold risk of PE (OR 15.2 [4.0–76.5], *p* = 0.0002).

### 3.5. Predictive Factors Associated with Prognosis

After adjustment of anticoagulant status and delay of follow-up, the composite criterion combining D-dimer level ≥ 2000 ng/mL and neutrophils count ≥ 7.0 G/L was associated with increased risk of death or ICU transfer (HR 12.9 [95%CI 2.5–67.8], Log-rank *p* < 0.01, Figure 2). This composite criterion was also associated with longer hospital stay (mean ± SD, 20 ± 6 vs. 14 ± 7 days, *p* = 0.01). The PE occurrence did not influence time to ICU transfer or death, nor did it influence time to hospital discharge (Log-rank *p* = 0.10 and 0.12 respectively; Appendix A
Appendix A).

## 4. Discussion

In the current study, we did analyze factors associated with the diagnosis of PE and with poor prognosis among a nationwide retrospective French cohort of non-ICU COVID-19 patients hospitalized in general wards. Our results indicate that, at the time of PE suspicion, the absence of anticoagulant agent (therapeutic long-term or prophylactic dose), a D-dimer level ≥ 2000 ng/mL and a neutrophil count ≥ 7.0 G/L were independent predictive factors of PE. In addition, fulfilling the composite criterion associating D-dimer ≥ 2000 ng/mL and neutrophils count ≥ 7.0 G/L was associated with a poor prognosis illustrated by a 12.9-fold risk of death or transfer to ICU.

The indication of CTPA is usually based on clinical suspicion. Here we showed that the input of the Wells score was very low in COVID-19 patients [15]. CTPA is probably advised for COVID-19 patients with limited disease extension requiring oxygen therapy or in patients who exhibit discrepancy between pulmonary lesions and the severity of respiratory failure [13,16]. Hence, an adapted risk stratification is essential; it should ideally take into account some demographic characteristics, level of D-dimer or other biomarkers, the risk of drug interaction, and risk of major bleeding [17,18]. In medical wards, the incidence of PE ranges from 5 to 10% [7,19,20,21]. Despite that around 50% of thromboembolic event occurs during the COVID-19 onset or upon admission [19], some of these events can occur during hospitalization [20,22].

As previously reported, we found low proportion of patients with PE who presented another thrombotic risk factor [7,15,19] nor any clinical features except the lack of anticoagulation [9,15,23]. The latter may suggest in situ pulmonary thrombosis partly mediated by vascular damage and thrombo-inflammation pathogenesis related to severe COVID-19 rather than pulmonary emboli [15]. Implication of neutrophils in these processes could sustain this hypothesis. The lack of deep venous thrombosis screening in the current cohort did not allow us to confirm this hypothesis. Moreover, in accordance with the absence of thrombocytopenia, normal PT ratio, and a high fibrinogen level, no disseminated intravascular coagulopathy (DIC) was suspected in our patients [24]. Several frequent clinical situations associated with venous thromboembolism [25], such as hormonal therapy or surgery, were barely or even not found in this and previous COVID-19 patients cohorts [26].

As expected, the lack of any long term anticoagulation or thromboprophylaxis was a risk factor associated with the occurrence of PE, in accordance with previous reports [21,22,23]. More than three quarters of PE can occur under thromboprophylaxis, possibly due to an insufficient prophylaxis dose [27,28]. Patients who need to be admitted to hospital should have a prescription for thromboprophylaxis unless contraindicated, but the best effective dosage remains uncertain [29]. A recent meta-analysis suggested better primary prevention of VTE by using curative versus prophylactic anticoagulation [30]. However, the expected benefit on survival following curative versus prophylactic dosing regimen (standard vs. high) still remains controversial [17,31]. Randomized clinical trials (RCTs) are ongoing to provide answers to this issue (e.g., ClinicalTrial number: *NCT04600141, NCT04344756*). The benefit of anticoagulation may also be related to non-anticoagulant properties, such as the decrease of plasma IL-6 levels and endothelial lesions, in association with better viral replication control and decreased risk of cytokine storm [24,32,33].

In addition of the lack of anticoagulation, D-dimer level was independently identified as predictive to PE. In the COVID-19, the best D-dimer cut-off to predict the occurrence of PE is variable across studies, depending on the studied population (upon admission vs. during hospitalization, general ward vs. ICU). The cut-off range has varied between 1000 and 5000 ng/mL in the literature [7,15,21,27]. Confounding factors, including previous steroid treatment duration, other immunomodulatory therapy, antibiotics, or anticoagulants, might influence the evolution of biomarkers levels and explains our choice to separate the upon admission vs. post admission population. Two authors with similar populations compared to the current cohort found similar a cut-off value to best predict occurrence of PE (2590 and 2247 ng/mL, respectively), with sensitivity ranging between 72 and 83%, specificity ranging between 74 to 83%, NPV ranging between 90 and 100%, PPV ranging between 48 and 73%, and an AUC ranging between 0.75 to 0.88, consistent with our findings [23,34,35]. As observed by others, the initial D-dimer level upon admission was independently predictive of the occurrence of PE [7]. Its monitoring over time may be relevant to prognosis staging and deserve further evaluation in future studies.

Both white blood cell and neutrophil counts were higher in PE group, consistent with previously published findings [22,23]. Many neutrophil activation biomarkers (e.g., Neutrophil Extracellular Traps (NETs), release of matrix metalloproteinase-9) were previously correlated with severity of COVID-19 infection and deep venous thrombosis or PE occurrence. These may contribute to immunothrombosis and the prothrombotic state in COVID-19 [36,37]. Interestingly, the high density of NETs formation containing microthrombi enriched of neutrophils and platelets found during COVID-19 is blocked by therapeutic dose of heparin [37]. Many other biomarkers have been described as associated with the severity of COVID-19 or the risk of thrombosis, such as inflammatory markers (e.g., *interleukin*-6) or hormonal markers [38,39,40].

Regarding prognosis, many studies have shown an impact of PE on death or the requirement of mechanical ventilation [7,20,21]. As previously described, we also found higher rate of ICU transfer in PE group but no difference on the PE-related mortality [15,22,27]. The non-critically ill status and the predominant unilateral and peripheral location of PE may explain these results [19,21,28]. In addition, anticoagulation was independently associated to lower mortality, although theses previous findings came from retrospective studies [41,42,43,44]. A strong association between high D-dimer levels and worse prognosis was established in COVID-19 patients, with a level 3.5-fold higher in non-survivors. A threshold >1000 ng/mL upon admission was associated with a 18-fold higher in-hospital mortality [43,45]. An increase of neutrophils count was also associated with higher mortality [46,47] and is also usually associated with high D-dimer level with a pro-coagulant state [48,49]. This may support the utility of combining these variables in the assessment of thromboembolic risk. Indeed, interaction between coagulation and neutrophils is also emphasized by higher levels of initial and peak of D-dimer levels and neutrophil-lymphocyte count ratio in deceased patients as compared to survivors [50]. Our findings highlighted an association between neutrophils and D-dimers and risk of death or ICU transfer.

This work has some limitations. The retrospective design can promote some bias and hidden confounders, we had some missing data inherent to automatically extracted information from clinical records, and the small sample size may have probably decreased our statistical power. The current cohort was, however, built on consecutive cases of confirmed PE, excluding patients with deep venous thrombosis and splitting the population between upon and post admission, hence overcoming the differences in thromboprophylaxis and specific treatment prescribed during hospitalization for homogeneity purpose in order to control variability, which in turn may have slightly increased statistical power despite the small sample size. This design choice was decided, consistent with the real-life setting of our study, in order to avoid an imputation of missing data due to the violation of the missing at random assumption. Our patient sample is homogeneous as compared to other published cohorts about VTE occurring on COVID-19 illness in patients hospitalized in general wards [7,15,19]. The heterogeneity of identified predictors of PE across previously published studies sustain the need of integrative multi-level diagnostic process. Further prospective validation of these potential diagnostic and prognostic predictive factors is warranted. But the short timeframe for data collection makes it possible to capture practices over a period limiting this variability.

## 5. Conclusions

Severe non-ICU COVID-19 patients hospitalized in medicine wards who presented with a D-dimer level ≥ 2000 ng/mL or a neutrophils count ≥ 7.0 G/L or absence of anticoagulant agent may be at risk of developing PE and should be therefore considered as candidates for a CTPA by prescribing physicians. The composite criterion D-dimers ≥ 2000 ng/mL plus neutrophils ≥ 7.0 G/L is identified as a predictive factor for a poor prognosis characterized by higher risk of death or ICU transfer, further emphasizing the need of careful monitoring of these patients in the clinical setting in order to continue improving the management of COVID-19 infection and hopefully decrease in-hospital mortality of this emerging viral infection.

## Figures and Tables

**Figure 1 viruses-13-00758-f001:**
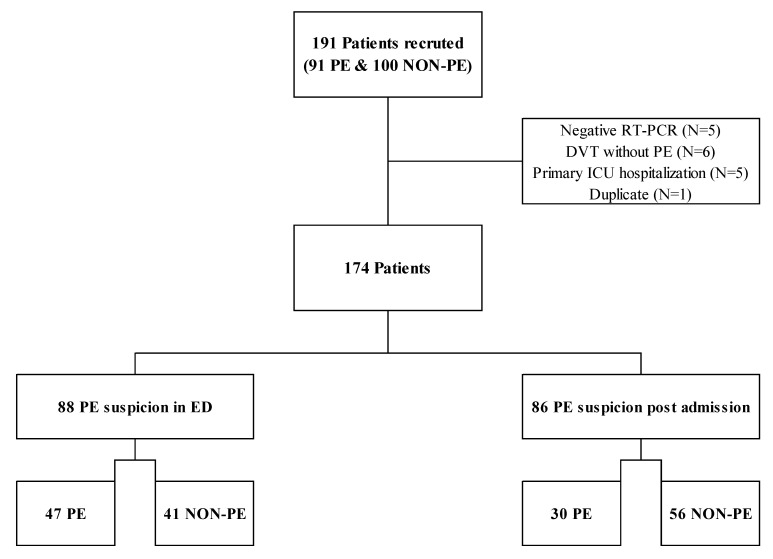
CLOTVID study flowchart diagram for patients hospitalized in medicine wards and with pulmonary embolism suspicion ≥ 24 h after admission. DVT, deep venous thrombosis; ED, emergency department; ICU, intensive care unit; PE, pulmonary embolism; NON-PE, absence of PE; RT-PCR, reverse transcriptase-polymerase chain reaction.

**Figure 2 viruses-13-00758-f002:**
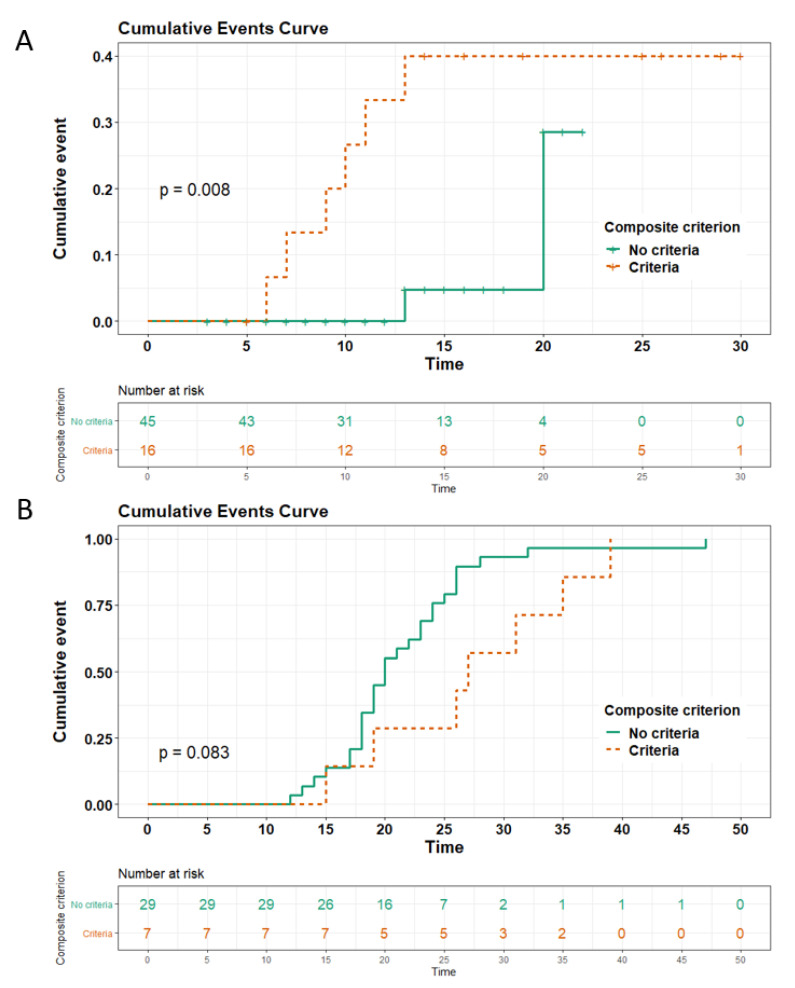
Cumulative events curve of ICU transfer or death in panel (**A**) and time to hospitalization discharge in panel (**B**), depending on the presence of double criteria D-dimers ≥ 2000 ng/mL and neutrophils count ≥ 7 G/L (composite outcome). The *p* represents the global *p*-value of Log-rank test in the Cox model. The risk table represents the patients still at risk of the event.

**Table 1 viruses-13-00758-t001:** Baseline characteristics of population at the Pulmonary Embolism suspicion.

	All *N* = 86	PE *N* = 30	NON-PE *N* = 56	*p*-Value
Baseline Characteristics				
Age [years]	66 [55–77]	64 [56–74]	68 [55–77]	0.5990
Age ≥ 65 years, n(%)	45 (52.3)	12 (50.0)	30 (53.6)	0.7519
Sex gender [male], n(%)	50 (58.1)	23 (76.7)	27 (48.2)	0.0108
Body mass index [kg/m^2^]	27.6 [24.2–21.2]	25.9 [23.9–30.0]	27.7 [25.0–32.1]	0.2921
Comorbidities				
Number of comorbidities by each patient	1 [1,2]	1 [0–2]	1 [1–2]	0.1431
Respiratory disease, n(%)	12 (14.0)	3 (10.0)	9 (13.1)	0.5292
Obesity (BMI > 30 kg/m^2^), n(%)	19 (22.1)	5 (16.7)	14 (25.0)	0.4267
Arterial hypertension, n(%)	44 (51.2)	12 (40.0)	32 (57.1)	0.1295
Diabetes mellitus, n(%)	24 (27.9)	10 (33.3)	14 (25.0)	0.4553
Cardiovascular disease, n(%)	19 (22.1)	7 (23.3)	12 (21.4)	1
Active smoking, n(%)	5 (5.8)	1 (3.3)	4 (7.1)	0.6537
Immunodeficiency, n(%)	4 (4.7)	0	4 (7.1)	0.2929
CTD or systemic vasculitis, n(%)	10 (11.6)	2 (6.7)	8 (14.3)	0.4827
Thrombo-embolic risk factor:				
Presence of at least one risk factor of venous thrombosis, n(%)	11 (12.8)	3 (10.0)	8 (14.3)	0.7402
Long term curative anticoagulant therapy, n(%)	11 (12.8)	3 (10.0)	8 (14.3)	0.7402
COVID-19 History				
Time from first attributable symptoms to hospital admission (days)	7 [5–10]	7 [5–10]	7 [5–10]	0.8770
Symptoms at admission:				
Fever > 38 °Celsius, n(%)	80 (93.0)	28 (93.3)	52 (92.9)	0.8204
Cough, n(%)	52 (60.5)	20 (66.7)	32 (57.1)	0.4430
Dyspnea, n(%)	62 (72.1)	24 (80.0)	38 (67.9)	0.2793
COVID-19 pneumonia HRTC:				
Ground glass, n(%)	75 (87.2)	25 (83.3)	50 (89.3)	0.1627
Consolidation, n(%)	53 (61.6)	15 (50.0)	38 (67.9)	0.0993
Absence or minimal extension, n(%)	17 (19.8)	5 (16.7)	12 (21.4)	1
Moderate extension, n(%)	36 (41.9)	11 (36.7)	25 (44.6)	0.8128
Extensive and severe, n(%)	27 (31.4)	10 (33.3)	17 (30.4)	0.6164
Time from first attributable COVID-19 symptoms to CTPA [days]	15 [10–18]	15 [12–21]	15 [10–17]	0.3638
Vital parameters at the time of CTPA:				
Heart rate/min	93 [79–105]	97 [81–105]	90 [73–106]	0.7994
Respiratory frequency [/min]	29 [23–35]	31 [23–39]	29 [24–33]	0.5150
Respiratory frequency > 22/min, n(%)	41 (47.7)	17 (56.7)	24 (42.9)	0.9272
SpO2 [%]	96 [93–97]	96 [93–97]	95 [93–96]	0.1720
SpO2 < 96%, n(%)	32 (37.2)	10 (33.3)	22 (39.3)	0.0943
Oxygen [Liter/min], mean ± SD	7 ± 12	5 ± 5	8 ± 15	0.2585
Oxygen flow ≥ 6 L/min, n(%)	35 (40.7)	14 (46.7)	21 (37.5)	1
Laboratory parameters:				
Platelets [G/L]	307 [231–389]	303 [253–331]	313 [221–431]	0.7106
WBC [G/L]	7.4 [4.9–9.6]	8.9 [6.7–11.9]	6.7 [4.7–8.4]	0.0021
WBC < 4 G/L, n(%)	9 (10.5)	0	9 (16.1)	0.0249
WBC > 10 G/L, n(%)	20 (23.3)	11 (36.7)	9 (16.1)	0.0291
Lymphocytes [G/L]	1.0 [0.7–1.4]	1.1 [0.8–1.4]	0.9 [0.7–1.4]	0.2630
Neutrophils [G/L]	5.6 [3.4–7.6]	7.0 [4.4–9.5]	5.4 [3.0–6.8]	0.0369
Neutrophils/Lymphocytes count ratio	5.6 [2.9–9.9]	6.8 [3.6–10.7]	4.8 [2.8–9.0]	0.4274
C Reactive protein, [mg/L]	96 [43–171]	77 [43–137]	102 [46–205]	0.4083
Ferritin [µg/L]	918 [524–1954]	957 [618–2233]	905 [524–1872]	0.6306
BNP [pg/mL]	77 [10–253]	65 [23–463]	86 [10–202]	0.6800
BNP level > 1500 pg/mL, n(%)	4 (4.7)	2 (6.7)	2 (3.6)	1
Troponin [ng/L]	9.0 [4.0–25.0]	12.0 [4.5–34.0]	8.0 [2.5–19.0]	0.3258
Serum creatinine [µmol/L]	70 [57–91]	79 [59–101]	68 [56–89]	0.5750
Prothrombin time ratio [%]	85 [75–91]	83 [75–86]	86 [75–94]	0.3077
Fibrinogen [g/L]	6.5 [4.9–7.9]	6.2 [4.7–7.5]	6.6 [5.1–7.9]	0.4774
D-dimer [ng/mL]	2678 [1460–8450]	9710 [3310–20,000]	1580 [863–2972]	<0.0001
D-dimer level ≥ 2000 ng/mL, n(%)	40 (46.5)	23 (76.2)	17 (30.4)	<0.0001

Data are presented in total (percentage) for categorical variables and median (interquartile range (IQR)) for continuous variables. BMI, body mass index; BNP, brain natriuretic peptide; Cardiovascular diseases: myocardial ischemia, cardiac injury, stroke; CTD, connective tissue disease; immunodeficiency: primitive or secondary immunodeficiency (CD4+T-cell < 0.2 G/L, ongoing chemotherapy, long-term steroids, or immunosuppressive therapy); PE, pulmonary embolism; respiratory disease: asthma, chronic obstructive pneumonia disease (COPD), chronic infiltrative pneumonia, etc.; CTPA, computed tomography pulmonary with angiography; HRCT, high-resolution computed tomography; WBC, white blood count; Statistical analyses for categorical variables by Chi2-test or exact Fisher test; and for quantitative variables by Mann–Whitney U-test. *p*-value: PE group vs. NON-PE group.

**Table 2 viruses-13-00758-t002:** Therapeutic Management and Outcomes.

	All *N* = 86	PE *N* = 30	NON-PE *N* = 56	*p*-Value
Treatment				
Standard of Care				
Oxygenotherapy, n(%)	170 (81.4)	21 (70.0)	49 (87.5)	0.0785
Ventilation:				
Invasive mechanical ventilation, n(%)	2 (2.3)	2 (6.7)	0	0.1190
Optiflow, n(%)	4 (4.7)	1 (3.3)	3 (5.4)	1
CPAP, n(%)	3 (3.5)	3 (10.0)	0	0.0396
Specific therapies				
Antiviral therapy, n(%)	24 (27.9)	7 (23.3)	17 (30.4)	0.2663
Immunomodulatory molecule, n(%)	25 (29.1)	9 (30.0)	16 (28.6)	1
Anti-IL6 receptor antibody, n(%)	8 (9.3)	2 (6.7)	6 (10.7)	0.7065
Steroids, n(%)	16 (18.6)	7 (23.3)	9 (16.1)	0.5625
Anticoagulation				
All anticoagulant agents, n(%)	78 (90.7)	24 (80.0)	54 (96.4)	0.0194
Previous long-term anticoagulation, n(%)	11 (12.8)	3 (10.0)	8 (14.3)	0.7402
Presence of thromboprophylaxis ^a^, n(%)	71 (82.6)	24 (80.0)	47 (83.9)	0.7670
Standard prophylactic dose, n(%) *	58 (81.7)	20 (83.3)	38 (80.9)	1
High prophylactic dose, n(%) *	13 (18.3)	4 (16.7)	9 (19.1)	1
No thromboprophylaxis, n(%)	8 (9.3)	6 (20.0)	2 (3.6)	0.0194
PE Treatment at Acute Phase				
Anticoagulation at therapeutic dose, n(%)	30 (34.8)	30 (100)	-	NA
LMWH, n(%)	22 (25.6)	22 (73.3)	-	-
Unfractionated heparin, n(%)	6 (7.0)	6 (20.0)	-	-
Direct oral anticoagulant, n(%)	2 (2.4)	2 (6.7)	-	-
Outcome				
Follow-up period [days]	20 [16–26]	20 [16–30]	19 [17–24]	0.3594
Length of hospital stay [days]	13 [10–18]	15 [10–20]	12 [10–42]	0.2746
Death or ICU transfer, n(%)	12 (14.0)	7 (23.3)	5 (8.9)	0.1007
ICU transfer, n(%)	7 (8.1)	6 (20.0)	1 (1.8)	0.0065
Death, n(%)	7 (8.1)	3 (10.0)	4 (7.1)	0.6907
Hospital discharge, n(%)	52 (60.5)	14 (46.7)	38 (67.9)	0.0554

Data are presented in total (percentage) for categorical variables and median (interquartile range (IQR)) for continuous variables. Antiviral therapy comprised association ritonavir/lopinavir and hydroxychoroquin/chloroquin; PE, pulmonary embolism; CPAP, continuous positive airway pressure; ICU, intensive care unit; LMWH, low molecular weight heparin; Follow-up period represented the time between COVID19 first symptom and last medical visit. Length of hospital stay was calculated on patients alive and not transferred in ICU department; Statistical analyses for categorical variables by Chi2-test or exact Fisher test; and for quantitative variables by Mann–Whitney U-test; not applicable (NA); ^a^ Thromboprophylaxis was analyzed specifically in patients without long term curative anticoagulant therapy. * Proportions were calculated only on patients with thromboprophylaxis (i.e., on 24 in PE group and on 54 in NON-PE group). *p*-value: PE group vs. NON-PE group.

**Table 3 viruses-13-00758-t003:** Predictive factors of pulmonary embolism during the hospitalization in medicine ward.

Variable	Univariate Analysis			Multivariate Analysis		
	OR	95%CI	*p*-Value	Adjusted OR	95%CI	*p*-Value
Male gender	3.5	1.4–10.1	0.0130	-	-	-
No anticoagulant agent	6.8	1.4–48.4	0.0251	72.3	3.6–4384.8	0.0143
D-dimer ≥ 2000 ng/mL	15.6	3.9–105.5	0.0006	26.3	4.1–537.8	0.0041
Neutrophil count ≥ 7 G/L	3.7	1.4–10.2	0.0100	5.8	1.4–29.5	0.0214

Anticoagulant agent corresponded to long term curative anticoagulant therapy or thromboprophylaxis during the hospitalization.

**Table 4 viruses-13-00758-t004:** Test characteristics of biomarkers identified in multivariate regression model for predicting the PE diagnosis and of Wells’ score.

Parameters	Cut-Off Point	Cut-Off Point	
Threshold	D-Dimer Level 2000 ng/mL	Neutrophils Count 7 G/L	Wells’ Score Binary
Subject reached value of cut-off point, n(%)	40 (46.5)	26 (30.2)	13 (15.1)
Area under curve (95%CI)	0.87 (0.78–0.95)	0.64 (0.51–0.78)	0.53 (0.42–0.64)
Sensitivity (95%CI)	0.92 (0.74–0.99)	0.52 (0.32–0.71)	0.13 (0.04–0.31)
Specificity (95%CI)	0.57 (0.41–0.73)	0.77 (0.64–0.88)	0.84 (0.72–0.92)
Positive predictive value (95%CI)	0.57 (0.41–0.73)	0.54 (0.33–0.73)	0.31 (0.09–0.61)
Negative predictive value (95%CI)	0.92 (0.74–0.99)	0.76 (0.62–0.87)	0.64 (0.52–0.75)
Positive likelihood ratio (95%CI)	2.16 (1.48–3.16)	2.29 (1.24–4.24)	0.83 (0.28–2.47)
Negative likelihood ratio (95%CI)	0.14 (0.04–0.54)	0.62 (0.41–0.95)	1.03 (0.86–1.24)

## Data Availability

Anonymized patient data were collected by local investigators by means of a standardized clinical report form (CRF) and then centralized by the three principal investigators (JG, BT and DS).

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
