# Peer review of "D-Dimer Level and Neutrophils Count as Predictive and Prognostic Factors of Pulmonary Embolism in Severe Non-ICU COVID-19 Patients"

_viruses, 2021, doi:10.3390/v13050758_

Round 1

Reviewer 1 Report

I read with interest the manuscript by Dr. Thoreau and Colleagues, aimed to identify clinical, biological and CT-scan predictive factors of PE in hospitalized non-ICU COVID-19 patients with suspicion of PE occurring >24 hours after their transfer from ED to medicine wards. This is part of the French CLOTVID retrospective cohort.

The manuscript is well written, English language is appropriate as well as the statistical analysis.

Minor issues:

-in “Patients and inclusion criteria” Authors stated “CT pulmonary angiography (CTPA) was performed for clinical suspicion of PE”. The clinical suspicion seems a subjective and personal criterion. Did the Authors identify a priori criteria for the clinical suspicion? If not, please highlight this issue in Methods and in Discussion.

-in “Data collection” Authors stated “The collected data were previously described (under review)”. Unfortunately at this stage that paper is not available, so please specify these data in detail.

-in “Patients’ characteristics” section, Authors stated “In PE patients, thrombus localization was proximal in 6 patients (20%), purely segmental in 19 (63.3%) …”. What about other five patients?

-Why 8 (6 PE and 2 NON-PE) of 86 patients of the PE suspicion post admission group did not receive thromboprophylaxis?

Reviewer 2 Report

Sir, 

I have recently reviewed the manuscript "D-dimer level and Neutrophils count as Predictive and Prognostic factors of Pulmonary Embolism in severe non-ICU COVID-19 patients" submitted by Benjamin Thoreau and co-workers to Viruses (MDPI). 

I can gladly say that this is an interesting work with a reasonable number of enrolled individuals. It is a very practically oriented manuscript and offers an interesting insight into a potentially life-threatening complication in Covid-19 patients. 

Obviously, the statistical analysis seems to be robustly and properly planned. It is greatly important. However, it is somewhat surprising that authors feel the need to include trivial graph explanations (e.g. in Fig. 2, I quote:  ... Data beyond the end of the whiskers are called "outlying" points and are plotted individually. Each dot representing one observation.). I believe it is not necessary. More importantly, it seems that some data exceeded the scale for the measurement of D-dimers, right? This aspect of statistical evaluation seems to be more puzzling. But this is acceptable as this is not an experimental work and all data are coming from routine diagnostic facilities (not oriented on scientific 

The authors included also patients using anti-IL6 therapy. This is an immensely interesting topic. Were actual IL6 levels measured/available at that point? IL-6 seems to be a critically important molecule in the course of Covid-19 disease. This might be a potential biomarker (please see recent review: Brábek J, et al.  Interleukin-6: Molecule in the Intersection of Cancer, Ageing and COVID-19. Int J Mol Sci. 2020 Oct 26;21(21):7937. doi: 10.3390/ijms21217937. PMID: 33114676; PMCID: PMC7662856.). This should be also properly discussed. 

I believe that authors should also pay attention to the influence of hormones - (as I see the cohort - at this age group potentially hormone replacement therapy). I do not see any comment on this aspect. However, several papers had raised concerns about the safety profile of HRT (including embolism risk), it should be discussed (see BMJ 2019364 doi: https://doi.org/10.1136/bmj.k4810 )

Also, the graphical value of Figure 3 is troublesome. The legibility here is suboptimal (contrasting to other Figures in this manuscript which are OK). Please, improve it. Also, the colours used are not very contrasting (at least for me). 

To conclude, I believe that this manuscript is worthy of my support. I believe that a few suggested changes can increase the value of this otherwise interesting work and authors can do these corrections in virtually no time. I am keen to reevaluate their work in the nearest future and approve for acceptance consequently. 

Reviewer 3 Report

Here are some critical remarks for the manuscript:
1. Title: why "Running Title"?
2. Are "Highlights" required in Viruses?
3. L69-71 - This aspect should be described in more detail. It should be emphasized (which the authors indirectly do) that the pathomechanism of coaguopathy in COVID-19 is multifactorial, but hypercoagulability and lack of fibrinolysis play a key role in its course (doi: 10.1055/a-1346-3178).
4. I am asking that authors always use the full name of disease (COVID-19).
5. The authors rightly note that D-dimer is still a controversial PE and DVT marker in the COVID-19 , which is of course a good starting point for undertaking these studies.
6. I have doubts whether it is worth quoting the work that is also in the review. It is worth considering an additional source.
7. Ferritin and WBC are inflammatory markers so maybe  write "D-dimer levels and laboratory markers of inflammatory responce (WBC count and ferritin levels)...?
8. The aim of the work is understandable but it should be more prominent.
9. Was the French CLOTVID study registered or were the early studies described somewhere?
10. The very short period of the research is puzzling - I am asking the authors for their comments.
11. The inclusion and exclusion criteria should be organized into a more logical description.
12. The clinical characteristics of the patients are very detailed and do not raise any objections. My only small note is the change from biological to laboratory parameters.
13. Figure 2 repeats the results from the table - do not need to be repeated.
14. The rest of the results and the discussion do not raise any objections, except for the necessity to check the English language in the discussion by a native speaker.

Round 2

Reviewer 3 Report

The authors have addressed all the comments of the reviewer and revised the manuscript accordingly.